# Trends in Outpatient Prescribing Patterns for Ocular Topical Anti-Infectives in Six Major Areas of China, 2013–2019

**DOI:** 10.3390/antibiotics10080916

**Published:** 2021-07-28

**Authors:** Zhenwei Yu, Jianping Zhu, Jiayi Jin, Lingyan Yu, Gang Han

**Affiliations:** 1Department of Pharmacy, Sir Run Run Shaw Hospital, Zhejiang University School of Medicine, Hangzhou 310016, China; yzw_srrsh@zju.edu.cn (Z.Y.); zjping@zju.edu.cn (J.Z.); 2Biomedical Research Center, Sir Run Run Shaw Hospital, Zhejiang University School of Medicine, Hangzhou 310016, China; 13587224234@163.com; 3Department of Pharmacy, Second Affiliated Hospital, Zhejiang University School of Medicine, Hangzhou 310009, China

**Keywords:** antibiotic, ophthalmologist, levofloxacin, tobramycin, prescription, cost

## Abstract

Topical anti-infectives are important in the management of ocular infections, but little is known about their current status and trends in their use in China. Thus, we carried out a prescription-based, cross-sectional study using the database of Hospital Prescription Analysis Projection of China, and aimed to analyze the trend in the use of ocular topical anti-infectives for outpatients of the ophthalmology department from 2013 to 2019. A total of 2,341,719 prescriptions from 61 hospitals located in six major areas written by ophthalmologists for outpatients were identified, and 1,002,254 of the prescriptions contained at least one anti-infective. The yearly anti-infective prescriptions increased continuously from 126,828 prescriptions in 2013 to 163,434 prescriptions in 2019. The cost also increased from 4,503,711 Chinese Yuan (CNY) in 2013 to CNY 5,860,945 in 2019. However, the use rate of anti-infectives decreased slightly from 46.5% in 2013 to 41.1% in 2019. Patients aged between 19 and 45 years old had the highest anti-infective use rate. Levofloxacin was the most frequently used anti-infective and kept on increasing among all age groups, occupying 67.1% of the total cost at the end of the study. Tobramycin was more frequently used in pediatric patients than in adults, but the use still decreased. Ganciclovir was the preferred anti-viral drug over acyclovir. In conclusion, the prescriptions and cost of ocular topical anti-infectives for outpatients both increased progressively. The increasingly widespread use of levofloxacin raised concerns regarding safety in pediatrics and resistance development. The observed trends can lead to the more efficient management of ocular anti-topical anti-infectives in China.

## 1. Introduction

Topical ocular anti-infectives played an important role in the management of ocular infections, including conjunctivitis, keratitis, blepharitis endophthalmitis and dacryocystitis [1,2]. Treatment failure of ocular infections may lead to vision loss and blindness [3]. Moreover, anti-infectives are also commonly used in perioperative prophylaxis with various patterns [4]. It was reported that the use of ocular topical anti-infectives increased in some countries [5]. This raised concerns regarding the over-use and inappropriate use of topical anti-infectives [6].

The inappropriate use of anti-infectives, especially the over-use of antibiotics, may lead to the development of antimicrobial drug resistance [7]. In this case, infection treatment would be more difficult, as many anti-infectives have no suitable formulation for ocular use. Moreover, inappropriate use, especially overuse, corresponds to a substantial cost. It is reported that the cost of post-cataract surgery antibiotic use in 2016 in the U.S. was USD 170 million, and there is substantial opportunity to improve the value of care [8]. Hospital antimicrobial stewardship programs had been implemented and the appropriateness of antibiotic use was improved in recent years [9]. However, ocular topical anti-infectives were not always included in these programs. The inappropriate use of antibiotic eye drops seems to be common among ophthalmologists in China [10]. Thus, better management of these drugs is urgently needed. A drug utilization study is an essential tool for evaluating and monitoring the prescribing patterns and improving the quality of pharmacological therapy.

For now, little is known about the current status and trend in the use of ocular topical anti-infectives in China, and it is essential for future stewardship programs to develop this understanding. Thus, we carried out this prescription-based, cross-sectional study to evaluate trends in the use of topical anti-infectives during a seven-year period.

## 2. Results

### 2.1. Total Trends in Prescriptions, Cost and Use Rate

A total of 2,341,719 prescriptions from 61 hospitals written by ophthalmologists for outpatients were identified, and 1,002,254 of the prescriptions contained at least one ocular topical anti-infective. As shown in Figure 1A, the yearly anti-infective prescriptions increased continuously from 126,828 prescriptions in 2013 to 163,434 prescriptions in 2019 (*p* = 0.007). The yearly expenditure on anti-infectives also increased progressively from 4,503,711 Chinese Yuan (CNY) in 2013 to CNY 5,860,945 in 2019 (*p* = 0.016). However, the use rate of anti-infectives (referred to as the percentage of prescriptions that contained ocular topical anti-infectives) showed a slight but significant decreasing trend (Figure 1B, *p* = 0.005).

### 2.2. Trends Stratified by Age and Sex

The demographic characteristics of patients who received anti-infective prescriptions are shown in Table 1. The numbers of anti-infective prescriptions for each age group all increased (all *p* < 0.05). Patients aged between 19 and 45 years old constituted the majority of patients who consumed ocular topical anti-infectives. The proportions of each age group did not change substantially during the study period. However, as shown in Figure 1B, the percentages of prescriptions containing anti-infectives for each age group differed greatly. The use rate of patients aged 65 years and over was stable during the study period, while the rate of pediatric patients decreased significantly (*p* = 0.006 for patients aged 2–8 years; and *p* = 0.004 for patients aged 9–18 years). Interestingly, patients aged between 19 and 45 years had the highest anti-infective use rate. Female patients were more common than males, about 1.3 times higher for each year, and but the ratios were not changed (*p* > 0.05). As shown in Figure 1C, monotherapy is the predominate pattern of topical anti-infective therapy, and its proportion kept on increasing during the seven-year period (*p* < 0.001).

### 2.3. Stratified by Drug

The yearly prescriptions of each specific drug and corresponding cost are shown in Table 2 and Table 3. Antibiotics were the main prescribed anti-infectives. Levofloxacin was the most frequently used topical inti-infective, and its use rate among anti-infective prescriptions continued to increase (*p* = 0.007). The cost of levofloxacin also increased (*p* = 0.016) and occupied 67.1% of the total cost of topical anti-infectives at the end of the study. Ofloxacin decreased both in prescription numbers and percentages (*p* = 0.007 and *p* < 0.001). Tobramycin showed a decreasing trend; however, its combination with dexamethasone increased progressively both in prescription numbers and portions (*p* = 0.003 and *p* < 0.001). We also analyzed the trends in use of levofloxacin and tobramycin in different age groups (Figure 2). The use of levofloxacin increased in all age groups (all *p* < 0.05). In contrast, tobramycin showed a decreasing trend in all age groups (all *p* < 0.05). However, the magnitude of the decrease in pediatric patients was smaller than that in adults, and still covered 32.7% of patients aged between 2 and 8 years old. The most frequently used anti-viral drug was ganciclovir, and anti-fungal drugs were rarely used.

## 3. Discussion

To the best of our knowledge, this was the first study that described the trends in ocular topical anti-infectives use in Chinese outpatients. As a large number of prescriptions from 61 sampling hospitals were included, the results are nationally representative. The prescription of ocular topical anti-infectives and the corresponding cost both increased during the seven-year period, but the percentage of prescriptions containing ocular topical anti-infectives declined. Antibiotics were the main anti-infectives, and levofloxacin was the favored anti-infective drug.

As the percentage of prescriptions containing ocular topical anti-infectives decreased, the increased number of prescriptions of topical anti-infectives may be due to increased demand and supply of ocular healthcare in China [11]. The reason for this decline may be the antimicrobial stewardship program in the ophthalmology departments. The corresponding cost was parallel to the increased prescriptions; thus, the health care system should be prepared to face increasing economic pressure. We noted that the cost dropped in 2018—this be may due to the implementation of the National Drug Centralized Procurement Program of China [12].

Significant differences in antibiotic use were found among countries. A population-based registry study was carried out to assess the ocular anti-infective use from 2015 to 2019 in a region of Spain. The results show that ocular topical anti-infectives were distributed to 5.38% of the population, and the use is increasing [5]. A study carried out in Demark, Norway and Sweden aimed to analyze ocular topical antibiotic use in children aged 0–4 years from 2000 to 2015. The incidence rate was stable between 2000 and 2010, and then declined [13]. It is reported that differences in healthcare systems, national guidelines and school and day care policies may contribute to the differences in antibiotic use [13]. Thus, improving these issues may lower the use of ocular anti-infectives.

Although there is a lack of direct comparisons with other countries, the approximately 40% prescription rate of anti-infectives in China raised concerns regarding the overuse of anti-infectives in ophthalmology departments. Topical anti-infectives are usually prescribed for ocular infection and perioperative prophylaxis. The most common ocular infection was reported to be conjunctivitis, and most cases of conjunctivitis are due to virus infection and are self-limiting, and thus, no antibiotic should be administered [6,14]. Even in the case of acute bacterial conjunctivitis, antibiotics are only recommended in severe cases [6]. Antibiotics were also found to be the most common medication prescribed for cataract surgery in some countries [15]. Nowadays, postoperative administration of topical antibiotics in noncomplicated cataract surgery, the most common clean surgery in ophthalmology, is a standard procedure that prevents and lowers endophthalmitis rates. However, some ophthalmologists use antibiotics in preoperative care, before the surgery, which is not proven to be beneficial for patients [16,17,18]. Moreover, in different ocular procedures, such as intravitreal injections, antibiotics administration is contraindicated [19]. The overuse of anti-infectives may be due to a lack of awareness of evidence or guidelines, institutionalized teaching or defensive practice [6].

When the anti-infective use was stratified by age, the highest use rate was found in patients aged between 19 and 45 years old, followed by patients aged between 2 and 8 years old. Thus, special attention should be paid to patients at these ages. It is not surprising that children had high anti-infective use, as the prevalence of bacterial ocular infection was high in patients at this age [14]. A higher use rate was also found in children in other countries [5]. However, the reason for the high use rate in adult patients is unknown.

Levofloxacin was the most frequently used antibiotic, and its use increased continuously during the study period. This situation was quite different from other countries. Tobramycin, including its combinations, is the most widely used anti-infective in Spain [5]. Chloramphenicol and fusidic acid were frequently used in pediatric patients in Denmark, Norway and Sweden [13]. Fluoroquinolones was frequently used for American post-surgery patients, but ofloxacin was the most common, while moxifloxacin and besifloxacin were the costliest antibiotics [15]. Tobramycin and fluoroquinolones are considered first-line drugs for the most common ocular infection, bacterial conjunctivitis [20]. Common organisms had similar susceptibility to levofloxacin and tobramycin, and these drugs had comparable efficacy in some ocular infections [21,22]. The continuously increasing use rate of levofloxacin should be noticed.

The systemic use of fluoroquinolones in children was warned against due to adverse drug reactions such as musculoskeletal events [23]. However, the safety of the topical use of fluoroquinolones in pediatric patients was controversial at one time [24]. Ophthalmologists were concerned about the safety of topical levofloxacin at the beginning of the study and tobramycin was frequently used in pediatric patients, and the use rate of levofloxacin in pediatric patients was much lower than that in adults. Recent studies considered that there was no evidence for harm caused by topical fluoroquinolone use in pediatrics [25,26,27,28]. Additionally, this might be the reason for the increasing use of levofloxacin in pediatric patients. However, the widespread use of levofloxacin raised concern regarding resistance development in common ocular pathogens [29].

Chloramphenicol and fusidic acid are also recognized as first-line treatments for some ocular infections; however, these drugs are seldom used in China. Chloramphenicol eye drops are cheap, have little irritative effect on the cornea and cover a broad spectrum of bacteria found in infections of the cornea or conjunctivae [30]. However, a probable association with aplastic anemia and high resistance in China are the reasons for its low use [22,30].

Ganciclovir was the most favorable ocular topical anti-viral drug in this study. However, it is more costly than acyclovir, which was the most widely used topical anti-infective in ocular viral infections in other countries [5]. There are a variety of ocular viral diseases caused by different types of viruses. Human adenovirus is the most common cause of infection on the ocular surface, accounting for up to 75% of conjunctivitis cases. Possible therapeutic benefits in adenovirus infection have been demonstrated through the use of antiviral drugs such as ganciclovir, ribavirin and cidofovir. The ophthalmic formulation of ganciclovir is used commonly in cases of adenovirus conjunctivitis and keratitis, which is superior to acyclovir [31,32,33]. However, ganciclovir was not superior to acyclovir in ocular human herpesvirus infection, another leading cause of infectious blindness [34]. Attention should be paid to the rational use of topical antiviral drugs. Anti-fungal drugs were seldom used, which was in accordance with the low prevalence of diagnosed ocular fungal infection [35]. Moreover, oral formulations and off-label use of injectable formulations are often used, but were not included for analysis in this study [36].

There were also some limitations of our research. Only prescriptions issued by ophthalmologists were included. The diagnosis was not included for analysis; thus, the appropriateness of topical anti-infective use was not evaluated. Most of the included hospitals were tertiary hospitals and sampling bias may exist.

## 4. Methods

### 4.1. Study Design and Data Source

This study was designed as a prescription-based, cross-sectional study. Ethical approval was obtained from the Ethical Committee of Sir Run Run Shaw Hospital, Zhejiang University School of Medicine (KEYAN20191011-18). Informed consent was waived as part of approval due to the nature of the study. The prescriptions were extracted from the database of the Hospital Prescription Analysis Cooperative Project. This database is widely used for pharmaco-epidemiology studies in China [37,38,39,40,41,42]. The database contains prescription information of participating hospitals on 40 random days each year. The prescription information covered prescription code, date, sex and age of patient, department of physician, drug name and cost.

### 4.2. Prescription Inclusion

Prescriptions meeting the following criteria were included for analysis: (1) prescriptions written by physicians of ophthalmology department; (2) prescriptions written for outpatients aged over 2 years during a study period from 2013 to 2019; (3) prescriptions from hospitals located in six major areas of China (Beijing, Shanghai, Guangzhou, Chengdu, Hangzhou, Tianjin) and hospitals that participated in the program continuously during the study period.

Topical drugs belonging to the S01A category and S01C category of ATC/DDD Index were defined as ocular topical anti-infectives [43]. Anti-infectives, including antibacterial drugs, anti-viral drugs and anti-fungal drugs, in the prescriptions were labeled for processing.

### 4.3. Data Analysis

The total yearly prescriptions and prescriptions containing topical anti-infectives written by ophthalmologists were counted. Then, the percentage of prescriptions that contained topical anti-infectives was calculated. The yearly expenditure on ocular topical anti-infectives was also added up. The yearly trends were analyzed and further stratified by age, sex and specific drug. The treatment pattern of ocular topical anti-infectives was also analyzed. The trends in yearly prescription numbers and cost were tested by the Mann–Kendal test, and the trends in percentages were tested by the log-linear test. All statistical tests were carried out in R V4.0.5 software.

## 5. Conclusions

Although the overall use rate declined during the study period, the number of prescriptions containing ocular topical anti-infectives increased progressively. Additionally, this raised concerns regarding inappropriate anti-infective use in outpatients of the ophthalmology department in China. The healthcare system may face the pressures of cost and the emergence of resistance. Ophthalmologists should follow the best clinical evidence to improve appropriate anti-infection therapy. Levofloxacin was prescribed most frequently, and its use increased among all age groups, including pediatric patients. However, resistance to levofloxacin among common pathogens causing ocular infections should be monitored closely. The observed trends can lead to the more efficient management of ocular anti-topical anti-infectives in China.

## Figures and Tables

**Figure 1 antibiotics-10-00916-f001:**
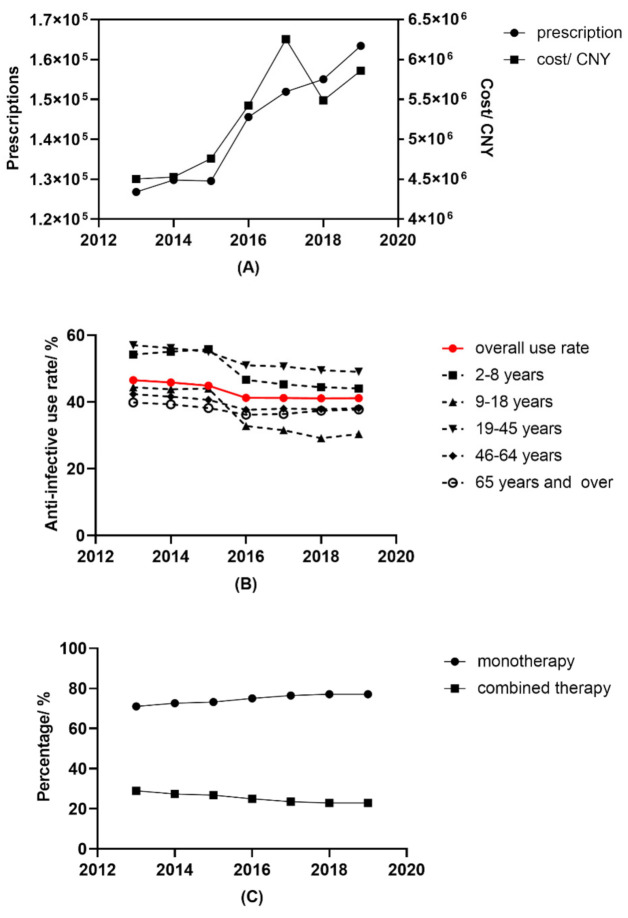
Trends in ocular topical anti-infective use in 61 sampling hospitals from 2013 and 2019. (**A**) Yearly prescriptions and cost; (**B**) overall use rate and age stratified use rate of ocular topical anti-infectives; (**C**) prescription patterns.

**Figure 2 antibiotics-10-00916-f002:**
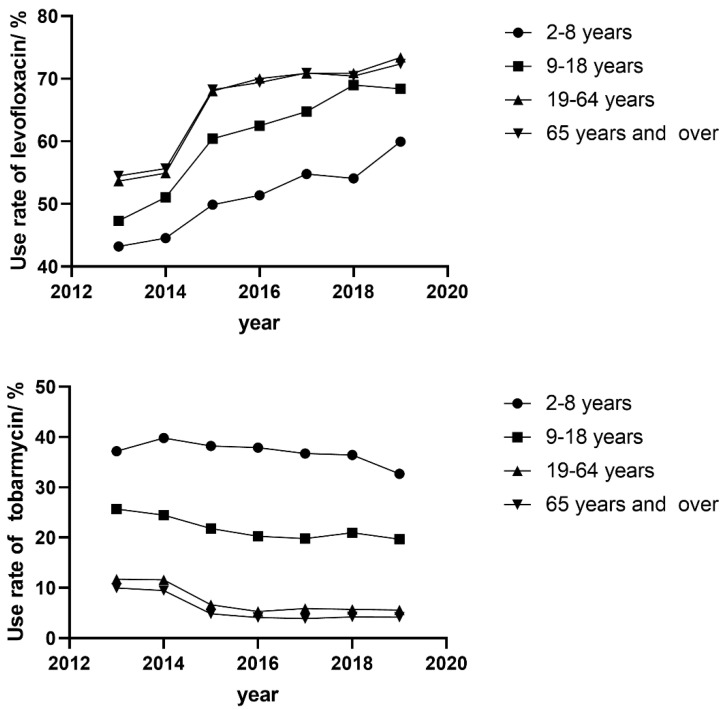
Trends in use rate of levofloxacin (**top**) and tobramycin (**bottom**).

**Table 1 antibiotics-10-00916-t001:** Demographic characteristics of study sample from 2013 to 2019 ^1^.

		2013	2014	2015	2016	2017	2018	2019
age	2–8 years	8424(6.64)	9252(7.13)	9703(7.49)	10,837(7.44)	11,139(7.33)	11,696(7.56)	12,720(7.78)
	9–18 years	5593(4.41)	5746(4.43)	5688(4.39)	6206(4.26)	6575(4.33)	6644(4.29)	7462(4.57)
	19–45 years	43,714(34.5)	43,959(33.9)	43,382(33.5)	49,683(34.1)	52,453(34.5)	53,615(34.6)	56,393(34.5)
	46–64 years	37,951(29.9)	38,426(29.6)	37,680(29.1)	42,131(28.9)	43,425(28.6)	42,872(27.7)	44,338(27.1)
	65 years and over	31,146(24.6)	32,421(25.0)	33,106(25.6)	36,760(25.2)	38,337(25.2)	39,956(25.8)	42,521(26.0)
sex	male	55,470(43.7)	56,530(43.6)	56,944(44.0)	63,803(43.8)	65,750(43.3)	66,418(42.8)	70,417(43.1)
	female	71,358(56.3)	73,274(56.4)	72,615(56.0)	81,814(56.2)	86,179(56.7)	88,665(57.2)	93,017(56.9)

^1^ Data were represented as prescriptions (percentage %).

**Table 2 antibiotics-10-00916-t002:** Numbers of ocular topical anti-infectives prescriptions from 2013 to 2019 ^1^.

Drugs	2013	2014	2015	2016	2017	2018	2019
Levofloxacin	67,077(52.9)	70,329(54.5)	86,020(66.4)	99,232(68.1)	105,462(69.4)	107,437(69.3)	117,406(71.8)
Ofloxacin	34,176(26.9)	33,068(25.6)	25,487(19.7)	27,503(18.9)	22,735(15.0)	21,651(14.0)	20,550(12.6)
Tobramycin/dexamethasone	17,187(13.6)	17,356(13.4)	18,500(14.3)	23,004(15.8)	23,950(15.8)	26,720(17.2)	28,564(17.5)
Tobramycin	17,248(13.6)	17,723(13.7)	11,947(9.22)	11,748(8.07)	12,536(8.25)	12,862(8.29)	13,039(7.98)
Erythromycin	5152(4.06)	5710(4.42)	4502(3.47)	4493(3.09)	5364(3.53)	5974(3.85)	6065(3.71)
Gatifloxacin	3839(3.03)	3470(2.69)	2274(1.76)	2180(1.5)	3274(2.15)	2947(1.9)	1917(1.17)
Lomefloxacin	2552(2.01)	1698(1.32)	1579(1.22)	921(0.63)	270(0.18)	161(0.1)	251(0.15)
Chlortetracycline	2088(1.65)	2019(1.56)	1535(1.18)	1419(0.97)	1726(1.14)	1850(1.19)	2036(1.25)
Sodium hyaluronate/chloramphenicol	1491(1.18)	1339(1.04)	1246(0.96)	934(0.64)	1269(0.84)	856(0.55)	1755(1.07)
Lincomycin	1233(0.97)	1130(0.88)	329(0.25)	480(0.33)	1127(0.74)	1027(0.66)	407(0.25)
Chloromycetin	1060(0.84)	1167(0.9)	1015(0.78)	683(0.47)	510(0.34)	346(0.22)	215(0.13)
Compound neomycin eye drops	951(0.75)	1118(0.87)	1049(0.81)	379(0.26)	105(0.07)	124(0.08)	42(0.03)
Rifampicin	573(0.45)	401(0.31)	113(0.09)	42(0.03)	10(0.01)	13(0.01)	9(0.01)
Tetracycline/cortisone	272(0.21)	365(0.28)	163(0.13)	-	-	-	-
Sulfacetamide sodium	153(0.12)	129(0.10)	121(0.09)	123(0.08)	77(0.05)	24(0.02)	-
Neomycin	93(0.07)	68(0.05)	38(0.03)	4(0.00)	15(0.01)	4(0.00)	-
Amikacin	51(0.04)	40(0.03)	9(0.01)	6(0.00)	9(0.01)	3(0.00)	-
Norfloxacin	50(0.04)	83(0.06)	44(0.03)	21(0.01)	1(0.00)	-	-
Fusidic acid	43(0.03)	42(0.03)	13(0.01)	18(0.01)	2(0.00)	-	-
Ciprofloxacin	22(0.02)	15(0.01)	2(0.00)	-	-	-	-
gentamicin/fluorometholone	-	-	-	-	282(0.19)	332(0.21)	205(0.13)
Ganciclovir	6831(5.39)	6512(5.04)	7093(5.47)	8139(5.59)	8761(5.77)	7953(5.13)	7669(4.69)
Acyclovir/sodium hyaluronate	2508(1.98)	2345(1.82)	2477(1.91)	2445(1.68)	2301(1.51)	2335(1.51)	2813(1.72)
Acyclovir	2729(2.15)	2508(1.94)	1606(1.24)	1005(0.69)	538(0.35)	412(0.27)	385(0.24)
Ribavirin	592(0.47)	474(0.37)	442(0.34)	309(0.21)	354(0.23)	299(0.19)	423(0.26)
Ftibamzone	44(0.03)	21(0.02)	-	-	-	-	-
Natamycin	16(0.01)	-	5(0.00)	44(0.03)	35(0.02)	28(0.02)	66(0.04)
Fluconazole	-	-	-	3(0.00)	-	1(0.00)	2(0.00)

^1^ Data are represented as prescriptions (percentage %).

**Table 3 antibiotics-10-00916-t003:** Expenditure on ocular topical anti-infectives dispensed from 2013 to 2019 ^1^.

Drug	2013	2014	2015	2016	2017	2018	2019
Levofloxacin	2,342,514(52.0)	2,453,344(54.2)	2,875,647(60.4)	3,372,923(62.2)	4,094,023(65.5)	3,575,850(65.2)	3,931,192(67.1)
Ofloxacin	749,390(16.6)	714,976(15.8)	640,714(13.5)	636,032(11.7)	544,397(8.70)	437,614(7.97)	419,011(7.15)
Tobramycin/dexamethasone	581,239(12.9)	587,307(13.0)	616,420(13.0)	749,588(13.8)	818,469(13.1)	804,675(14.7)	845,147(14.4)
Tobramycin	352,569(7.83)	352,601(7.79)	249,359(5.24)	248,459(4.58)	277,593(4.44)	235,096(4.28)	236,677(4.04)
Ganciclovir	215,112(4.78)	200,949(4.44)	208,749(4.39)	226,533(4.18)	230,249(3.68)	191,290(3.49)	182,981(3.12)
Other	262,887(5.84)	219,362(4.84)	168,587(3.54)	189,612(3.50)	290,198(4.64)	243,031(4.43)	245,938(4.20)

^1^ Data are represented as cost in Chinese Yuan (percentage %).

## Data Availability

Data is contained within the article.

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
