# Peer review of "Trends in Outpatient Prescribing Patterns for Ocular Topical Anti-Infectives in Six Major Areas of China, 2013–2019"

_antibiotics, 2021, doi:10.3390/antibiotics10080916_

Round 1
Reviewer 1 Report
The article is clearly presented but the English level needs major improvement starting from the abstract (e.g., "aimed to analysis" should be replaced by "aimed to analyze", and many more mistakes throughout the test should be corrected).
In the methods please make explicit the protocol number of Ethic Committee approval.
Author Response
The article is clearly presented but the English level needs major improvement starting from the abstract (e.g., "aimed to analysis" should be replaced by "aimed to analyze", and many more mistakes throughout the test should be corrected).
Response: The English writing has been checked by MDPI’s English editing service and we think the English in the revised manuscript is much better now.
In the methods please make explicit the protocol number of Ethic Committee approval.
Response: The protocol number of Ethic Committee approval is added as suggested.
Reviewer 2 Report
Thank you for allowing me to review this paper
- This is an epidemiological report of > 2millions cases- impressive
Intro:
- Inappropriate use of anti-infective leads to resistance. Is there evidence that this is the case of ocular topical anti-infective as well?
- My mentor would tell me that if one starts topical ABX before surgery and endophthalmitis occur later- it would be hard to treat. Still, I don’t know if there is any data to support lowering the use of topical anti-infective related to bacterial residence.
- I think a word about cost should also be in the introduction- could you relate to it as overuse of anti-infective might be a burden on the community.
- Section 2 is the results- does it come before methods? Is this the journal format?
Results
- Were you able to identify repeated prescriptions for the same patient? If a patient had an infection and he/she received two prescriptions (for drug a and drug b,) did you count this as 2 or 1?
- What happened if the same person received several prescriptions over the years?
- When you describe the trends of prescription given: it seems that in figure a, there is a steady rise, but in figure b, the overall use is not increasing – could you explain this?
- Could you explain the %overall use rate? What is this percentage mean?
- When you compare the percentage of year to year, you will get a statistical significance as this is a considerable number of cases – I don’t know the right way to compare yow might consider from consulting with a statistician.
- Why was there a drop in Cost/CNY in 2018? What is CNY?
- From table 1 – we can see that the rise is mostly in kids 2-8
- In table 1, what is the number in the () is it a percentage? If so, it looks like gender had a stable percentage while the numbers increased
- Levo vs. oflox? Are these not the same drugs?
- The acyclovir is topical, or did you also included tablets?
- For acyclovir, the trend is a decrease. Why is that, do you think?
- What is the difference between table 2 and table 3?
- Table 3: about the expenditure? What are the units- what is the () means?
- Discussions
- This is not clear to me: “The use ocular topical anti-infectives increased both in prescriptions and cost during the seven years, but the use rate declined” and “As the use rate of ocular topical anti-infectives decreased, the increased prescriptions may due to increased demand and supply of ocular health care in China” can you explain what is user rate and why it declined?
- This also not clear” the importance of antimicrobial stewardship program in the control of anti-infective use. “ what do you mean here? What is a stewardship program? Try to be more clinically oriented.
- Could you, in the intro/or here in discussion, review in-depth ref five from Spain and 18 from Denmark? How many patients? For how long? What was their outcome? It would benefit the reader may be to add a supp table or in the text.
- As you did not collect the diagnosis for the prescription, the discussion on conjunctivitis is speculative, and I think it should be removed/ rephrased.
- “High rate of anti-infectives prescribing” how do you know this high? What would be the average rate? Or standard rate?
- “But the reason of high use in adult patients should be investigated in future studies”- how would one investigate this? Any ideas?
- “Ophthalmologists were conservative at the beginning of study and tobramycin was frequently used in pediatric patients, and the use of levofloxacin in pediatric patients was much lower than that in adults.” – not sure “conservative” is the right word here as you did not conduct a survey on ophthalmologists opinions merely looked at prescription rate.
- There is no reference to cost in the discussion- as you reported cost in the results, you should discuss the implication of increased cost.
Methods:
- How did you compare the rate of increase? Along the entire period or 2019 vs. 2013?
- You collected prescriptions from the hospital. In Israel, a local pharmacist does prescriptions. What is the system in China?
- Did you use the “ log-linear test”? Do you need to use Kaplan-Meier for trend analysis?
- CONCLUSION: what is the clinical conclusion from this study? How will it benefit the patients? How will it benefit the medical system? How will it help the physician? Try to elaborate on this and give clinical relevance to this study.
Overall- I support the publication of this large data- please correct and send back with corrections. Please add a full answer under each comment without referring to the text.
Author Response
Review 2
Thank you for allowing me to review this paper
1. This is an epidemiological report of > 2millions cases- impressive
Intro:
2. Inappropriate use of anti-infective leads to resistance. Is there evidence that this is the case of ocular topical anti-infective as well?
Response: We have cited a reference here. The cited study concluded that the long-term use of eye drops containing benzalkonium chloride might select benzalkonium chloride -resistant S. epidermidis. So, we think it is suitable to write “The inappropriate use of anti-infectives, especially the over-use of antibiotics, may lead to the development of antimicrobial drug resistance.”.
3. My mentor would tell me that if one starts topical ABX before surgery and endophthalmitis occur later- it would be hard to treat. Still, I don’t know if there is any data to support lowering the use of topical anti-infective related to bacterial residence.
Response: There is many literatures reporting that microorganism exposure to antibiotic rapidly develop resistance. Thus, we think that lowering the use could decrease antibiotic exposure to bacteria and develop less resistance. We have revised the discussion about peri-operative antibiotic use as follows,
“Nowadays, postoperative administration of topical antibiotics in noncomplicated cataract surgery, the most common clean surgery in ophthalmology, is a standard procedure that prevents and lowers endophthalmitis rates. However, some ophthalmologists use antibiotics in preoperative care, before the surgery, which is not proven to be beneficial for patients.[16–18]”
4. I think a word about cost should also be in the introduction- could you relate to it as overuse of anti-infective might be a burden on the community.
Response: A brief introduction about cost by overuse of anti-infectives was added as suggest.
“Moreover, inappropriate use, especially overuse, corresponds to a substantial cost. It is reported that the cost of post-cataract surgery antibiotic use in 2016 in the US was USD 170 million, and there is substantial opportunity to improve the value of care.[8]”
5. Section 2 is the results- does it come before methods? Is this the journal format?
Response: Yes, it is the journal format.
Results
6. Were you able to identify repeated prescriptions for the same patient? If a patient had an infection and he/she received two prescriptions (for drug a and drug b,) did you count this as 2 or 1?
Response: Because the database is anonymous, we cannot identify repeated prescriptions for patients. This research is prescription based, and if a patient received two prescriptions, we count this as 2. We have declared this in the method. This situation is not common, the patients’ topical anti-infectives is usually written on a single prescription.
7. What happened if the same person received several prescriptions over the years?
Response: Because the database is anonymous, the specific patients can not be identified. If the same person received several prescriptions over the years, we only calculated the number of prescriptions.
8. When you describe the trends of prescription given: it seems that in figure a, there is a steady rise, but in figure b, the overall use is not increasing – could you explain this?
Response: We provide the possible explanation in the discussion section,
“As the percentage of prescriptions containing ocular topical anti-infectives decreased, the increased number of prescriptions of topical anti-infectives may be due to increased demand and supply of ocular health care in China.[11]”
9. Could you explain the %overall use rate? What is this percentage mean?
Response: We have added the explanation right after the term, referred to percentage of prescription that contained ocular topical anti-infectives
10. When you compare the percentage of year to year, you will get a statistical significance as this is a considerable number of cases – I don’t know the right way to compare yow might consider from consulting with a statistician.
Response: We used to test the significance in trends using log-linear test and it is suitable for the testing of trends in percentages over a seven-year period. Many previous literatures including ours used this method.
11. Why was there a drop in Cost/CNY in 2018? What is CNY?
Response: We have added the possible reason in the discussion. CNY is Chinese Yuan. We have provided the full spelling where the abbreviation first appeared. Reference was also cited.
“We noted that the cost dropped in 2018—this be may due to the implementation of the National Drug Centralized Procurement Program of China.[12]”.
12. From table 1 – we can see that the rise is mostly in kids 2-8
Response: Prescriptions for patients age 2-8 increased, but it is not the only group that increased. For example, patients age 65 years up increased from 24.6% to 26.0%.
13. In table 1, what is the number in the () is it a percentage? If so, it looks like gender had a stable percentage while the numbers increased
Response: We have added a footnote under the table to clarify the data expression. The gender had a stable percentage during the study period, and we have stated this in the result section.
“Data are represented as prescriptions (percentage).”
14. Levo vs. oflox? Are these not the same drugs?
Response: Levofloxacin is the levorotatory form of ofloxacin. They have similar pharmacological effects. But they have different ATC code (S01AE01 for ofloxacin and S01AE05 for levofloxacin). So, we treated them as different drugs.
15. The acyclovir is topical, or did you also included tablets?
Response: Acyclovir tablets are not included for analysis in this study. Only ocular topical anti-infectives were analyzed.
16. For acyclovir, the trend is a decrease. Why is that, do you think?
Response: The use of acyclovir decreased because of the increase of other anti-viral drugs, mainly ganciclovir. We have discussed the increased use of ganciclovir over acyclovir,
“Ganciclovir was the most favorable ocular topical anti-viral drug in this study. However, it is more costly than acyclovir, which was the most widely used topical anti-infective in ocular viral infections in other countries. [5] There are a variety of ocular viral diseases caused by different types of viruses. Human adenovirus is the most common cause of infection on the ocular surface, accounting for up to 75% of conjunctivitis cases. Possible therapeutic benefits in adenovirus infection have been demonstrated through the use of antiviral drugs such as ganciclovir, ribavirin, and cidofovir. The ophthalmic formulation of ganciclovir is used commonly in cases of adenovirus conjunctivitis and keratitis, which is superior to acyclovir.[31–33] However, ganciclovir was not superior to acyclovir in ocular human herpesvirus infection, another leading cause of infectious blindness. [34] Attention should be paid to the rational use of topical antiviral drugs.”.
17. What is the difference between table 2 and table 3?
Response: Table 2 showed the prescription numbers, and table 3 showed the cost of drugs.
18. Table 3: about the expenditure? What are the units- what is the () means?
Response: We have added a footnote to clarify the data meaning.
“Data are represented as cost in Chinese Yuan (percentage).”
19. Discussions
20. This is not clear to me: “The use ocular topical anti-infectives increased both in prescriptions and cost during the seven years, but the use rate declined” and “As the use rate of ocular topical anti-infectives decreased, the increased prescriptions may due to increased demand and supply of ocular health care in China” can you explain what is user rate and why it declined?
Response: The sentence is revised as follows to make it more clear,
“The prescription of ocular topical anti-infectives and the corresponding cost both increased during the seven-year period, but the percentage of prescription containing ocular topical anti-infectives declined.”
And
“As the percentage of prescriptions containing ocular topical anti-infectives decreased, the increased number of prescriptions of topical anti-infectives may be due to increased demand and supply of ocular health care in China.”
21. This also not clear” the importance of antimicrobial stewardship program in the control of anti-infective use. “ what do you mean here? What is a stewardship program? Try to be more clinically oriented.
Response: Thanks for the comments and we think the expression is inappropriate. We have revised the sentence as follows,
“The reason for this decline may be the antimicrobial stewardship program in the ophthalmology departments.”.
22. Could you, in the intro/or here in discussion, review in-depth ref five from Spain and 18 from Denmark? How many patients? For how long? What was their outcome? It would benefit the reader may be to add a supp table or in the text.
Response: We have added the details of these two research in the revised manuscript.
“A population-based registry study was carried out to assess the ocular anti-infective use from 2015 to 2019 in a region of Spain. The results show that ocular topical anti-infectives were dispended to 5.38% of the population, and the use is increasing. [5] A study carried out in Demark, Norway and Sweden aimed to analyze ocular topical antibiotic use in children aged 0-4 years from 2000 to 2015. The incidence rate was stable between 2000 and 2010, and then declined. [13]”
23. As you did not collect the diagnosis for the prescription, the discussion on conjunctivitis is speculative, and I think it should be removed/ rephrased.
Response: We have rephrased the expression as suggested.
“Topical anti-infectives are usually prescribed for ocular infection and perioperative prophylaxis. The most common ocular infection was reported to be conjunctivitis, and most cases of conjunctivitis are due to virus infection and are self-limiting, and thus no antibiotic should be administered.”
24. “High rate of anti-infectives prescribing” how do you know this high? What would be the average rate? Or standard rate?
Response: Thanks for your comments. There is no direct comparison to other countries, and it is inappropriate to conclude the rate of high. We have removed “high rates” and revised the sentence as follows,
“Although there is a lack of direct comparisons with other countries, the approximately 40% prescription rate of anti-infectives in China raised concerns regarding the overuse of anti-infectives in ophthalmology departments.”.
25. “But the reason of high use in adult patients should be investigated in future studies”- how would one investigate this? Any ideas?
Response: We have revised the sentence as follows,
“But the reason for high use in adult patients is unknown.”.
26. “Ophthalmologists were conservative at the beginning of study and tobramycin was frequently used in pediatric patients, and the use of levofloxacin in pediatric patients was much lower than that in adults.” – not sure “conservative” is the right word here as you did not conduct a survey on ophthalmologists opinions merely looked at prescription rate.
Response: Thanks for your comments. We have deleted the word “conservative”. Now the sentence is revised as follows,
“Ophthalmologists concerned about the safety of topical levofloxacin at the beginning of study and tobramycin was frequently used in pediatric patients”.
27. There is no reference to cost in the discussion- as you reported cost in the results, you should discuss the implication of increased cost.
Response: The discussion about increased cost is as follows,
“The corresponding cost was parallel to the increased prescriptions; thus, the health care system should be prepared to face increasing economic pressure. We noted that the cost dropped in 2018—this be may due to the implementation of the National Drug Centralized Procurement Program of China.[12]”.
Methods:
28. How did you compare the rate of increase? Along the entire period or 2019 vs. 2013?
Response: We test the trend in percentages along the entire period by log-linear test. Not 2019 vs 2013.
29. You collected prescriptions from the hospital. In Israel, a local pharmacist does prescriptions. What is the system in China?
Response: In China, anti-infecives, including ocular topical formulation, can be prescribed by doctors only. And nearly all patients get these drugs from hospital. Community pharmacy cannot sell ocular topical anti-infectives without prescription, so most pharmacies don’t put these drugs on shell.
30. Did you use the “log-linear test”? Do you need to use Kaplan-Meier for trend analysis?
Response: We use the log-linear test for trend in percentages of prescriptions that contained ocular topical anti-infectives. As this method is suitable, we haven’t considered other method such as Kaplan-Meier for trend.
31. CONCLUSION: what is the clinical conclusion from this study? How will it benefit the patients? How will it benefit the medical system? How will it help the physician? Try to elaborate on this and give clinical relevance to this study.
Response: Thanks for your comments. We have revised the conclusion as follows to give clinical relevance to this study.
“Although the overall use rate declined during the study period, the number of prescriptions containing ocular topical anti-infectives increased progressively. Additionally, this raised concerns regarding inappropriate anti-infective use in outpatient of ophthalmology department in China. The health care system may face the pressures of cost and the emergence of resistance. Ophthalmologists should follow the best clinical evidence to improve appropriate anti-infection therapy. Levofloxacin was prescribed most frequently, and its use increased among all age groups, including pediatric patients. However, resistance to levofloxacin among common pathogens causing ocular infections should be monitored closely. The observed trends can lead to the more efficient management of ocular anti-topical anti-infectives in China.”
Reviewer 3 Report
The authors brought a very urgent and important topic of global overuse, as well as inappropriate use of antibiotic eye drops among ophthalmologists.
They managed to evaluate the current status and trends in the antibacterial and antiviral topical medications use in China.
The article titled “Trends in outpatient prescribing patterns of ocular topical anti-infectives in six major areas of China, 2013-2019” present the perspective of topical anti-infectives usage among Chinese doctors in variety of ophthalmic infectious diseases. The subject of the study is up to date, because the knowledge regarding anti-infectives use is one of a global urgent and up to date topic, which should be discussed widely. The figures are properly made and the figure legends are clear for the reader.
However, as an ophthalmologist I have several issues, which have to be resolved:
- These sentences are not true in context of prophylaxis of endophthalmitis following cataract surgery.
“But antibiotic prophylaxis is not recommended in clean surgery.[6] High rate of anti-infectives prescribing may be due to lack of awareness of evidence or guidelines, institutionalized teaching or defensive practice. [6]”
The paper, the authors quote is not related to cataract surgery, but to lid surgery in cases like chalazion or blepharitis.
Evidence based medicine papers on prophylaxis in cataract surgery, for example:
Gower EW, Lindsley K, Tulenko SE, Nanji AA, Leyngold I, McDonnell PJ
Perioperative antibiotics for prevention of acute endophthalmitis after cataract surgery. Cochrane Database Syst Rev. 2017 Feb 13;2:CD006364.
Bandello F, Coassin M, Di Zazzo A, Rizzo S, Biagini I, Pozdeyeva N, Sinitsyn M, Verzin A, De Rosa P, Calabrò F, Avitabile T, Bonfiglio V, Fasce F, Barraquer R, Mateu JL, Kohnen T, Carnovali M, Malyugin B; Group LEADER-7 Investigators.
One week of levofloxacin plus dexamethasone eye drops for cataract surgery: an innovative and rational therapeutic strategy. Eye (Lond). 2020 Nov;34(11):2112-2122.
Aragona P, Postorino EI, Aragona E. Post-surgical management of cataract: Light and dark in the 2020s. Eur J Ophthalmol. 2020 Oct 20:1120672120963458.
Those papers, based mostly on clinical studies state clearly, that nowadays postoperative administration of topical antibiotics in noncomplicated cataract surgery is a standard procedure, that prevents and lowers endophthalmitis rates. However, some ophthalmologist use antibiotics also in the preoperative care, before the surgery, which is not proven to be beneficial for patients. In this situation, I fully agree, that it is an example of an overuse.
Also, in different ocular procedures, such as intravitreal injections antibiotics administration is contraindicated and this should be mentioned by authors. In this area, we actually meet a problem of overuse and inappropriate use of topical antibiotics.
Yin VT, Weisbrod DJ, Eng KT, Schwartz C, Kohly R, Mandelcorn E, et al. Antibiotic resistance of ocular surface flora with repeated use of a topical antibiotic after intravitreal injection. JAMA Ophthalmol. 2013;131:456–61.
Michelle EW, Adrienne WS. How to give intravitreal injections. 2013. https://www.aao.org/eyenet/article/how-to-give-intravitre al-injections
- “However, the safety of topical use of fluoroquinolones in pediatric patients are controversial.[25]”
Topical moxifloxacin, ofloxacin, ciprofloxacin and lexofloxacin are safe to be used in children older that 1 year old according to the summary of product characteristics in many countries, as in 2021.
Why in your opinion, the prescription of these drugs in children population is controversial?
- “Moreover, advantage of ganciclovir compared to acyclovir was not proved.”
There is a variety of ocular viral diseases caused by different types of viruses, such as adenoviruses, herpes simplex virus type-1, varicella zoster virus or cytomegalovirus.
As in human herpesviruses, there are also many forms of ocular involvement, such as conjunctivitis, blepharitis, keratitis, uveitis and retinitis. The estimated global incidence of HSV keratitis is roughly 1,5 million, including 40,000 new cases of each year. In HSV keratitis the treatment could be based on ocular and oral formulas of acyclovir or ganciclovir, without, as you wrote the advantage of ganciclovir compared to acyclovir.
However, it is not HSV, but human adenovirus (HAdV) the most common cause of infection to the ocular surface, accounting for up to 75% of conjunctivitis cases. Epidemic keratoconjunctivitis (EKC) is the most severe ocular form and is distinguished by its ability to invade the corneal epithelium, ranging in presentation from a keratitis to persistent and recurrent subepithelial infiltrates (SEIs). Possible therapeutic benefits in HAcV infection have been demonstrated through the use of antiviral drugs such as ganciclovir, ribavirin, and cidofovir.
The ophthalmic gel form of ganciclovir is used commonly in cases of HAcV conjunctivitis and keratitis. In case of this infection ganciclovir is superior to acyclovir.
Ying B, Tollefson AE, Spencer JF, et al. Ganciclovir inhibits human adenovirus replication and pathogenicity in permissive immunosuppressed Syrian hamsters. Antimicrob Agents Chemother. 2014;58(12):7171–7181. doi:10.1128/AAC.03860-14
Wold WS, Toth K. New drug on the horizon for treating adenovirus. Expert Opin Pharmacother. 2015;16(14):2095–2099. doi:10.1517/14656566.2015.1083975
Huang J, Kadonosono K, Uchio E. Antiadenoviral effects of ganciclovir in types inducing keratoconjunctivitis by quantitative polymerase chain reaction methods. Clin Ophthalmol. 2014;8:315–320. doi:10.2147/OPTH.S55284
- “Anti-fungal drugs were seldomly used, which was in accordance with the low prevalence of ocular fungal infection”.
The annual global incidence of fungal keratitis has never been estimated. In the recent paper published in 2020 in Lancet, the authors estimated that over a million eyes are affected each year from fungal keratitis—probably 1.4 million—assuming that in high-incidence areas culture-negative cases are usually cases of fungal keratitis. The fungal keratitis is most prevalent in tropical and subtropical locations and has been estimated to account even for 20–60% of all culture-positive corneal infections in these climates.
Thus, I am really surprised by such low use of antifungal medicines in Chine.
5. What about Acanthamoeba keratitis treatment? None in China?
- Weak sides. In Europe many topical anti-infectives prescriptions are issued by general doctors (family doctors, first-line doctors), not by ophthalmologist. This could be a reason of a significant bias in your research if the situation is similar in China.
Author Response
Reviewer 3
The authors brought a very urgent and important topic of global overuse, as well as inappropriate use of antibiotic eye drops among ophthalmologists.
They managed to evaluate the current status and trends in the antibacterial and antiviral topical medications use in China.
The article titled “Trends in outpatient prescribing patterns of ocular topical anti-infectives in six major areas of China, 2013-2019” present the perspective of topical anti-infectives usage among Chinese doctors in variety of ophthalmic infectious diseases. The subject of the study is up to date, because the knowledge regarding anti-infectives use is one of a global urgent and up to date topic, which should be discussed widely. The figures are properly made and the figure legends are clear for the reader.
However, as an ophthalmologist I have several issues, which have to be resolved:
1. These sentences are not true in context of prophylaxis of endophthalmitis following cataract surgery.
“But antibiotic prophylaxis is not recommended in clean surgery.[6] High rate of anti-infectives prescribing may be due to lack of awareness of evidence or guidelines, institutionalized teaching or defensive practice. [6]”
The paper, the authors quote is not related to cataract surgery, but to lid surgery in cases like chalazion or blepharitis.
Evidence based medicine papers on prophylaxis in cataract surgery, for example:
Gower EW, Lindsley K, Tulenko SE, Nanji AA, Leyngold I, McDonnell PJ
Perioperative antibiotics for prevention of acute endophthalmitis after cataract surgery. Cochrane Database Syst Rev. 2017 Feb 13;2:CD006364.
Bandello F, Coassin M, Di Zazzo A, Rizzo S, Biagini I, Pozdeyeva N, Sinitsyn M, Verzin A, De Rosa P, Calabrò F, Avitabile T, Bonfiglio V, Fasce F, Barraquer R, Mateu JL, Kohnen T, Carnovali M, Malyugin B; Group LEADER-7 Investigators.
One week of levofloxacin plus dexamethasone eye drops for cataract surgery: an innovative and rational therapeutic strategy. Eye (Lond). 2020 Nov;34(11):2112-2122.
Aragona P, Postorino EI, Aragona E. Post-surgical management of cataract: Light and dark in the 2020s. Eur J Ophthalmol. 2020 Oct 20:1120672120963458.
Those papers, based mostly on clinical studies state clearly, that nowadays postoperative administration of topical antibiotics in noncomplicated cataract surgery is a standard procedure, that prevents and lowers endophthalmitis rates. However, some ophthalmologist use antibiotics also in the preoperative care, before the surgery, which is not proven to be beneficial for patients. In this situation, I fully agree, that it is an example of an overuse.
Also, in different ocular procedures, such as intravitreal injections antibiotics administration is contraindicated and this should be mentioned by authors. In this area, we actually meet a problem of overuse and inappropriate use of topical antibiotics.
Yin VT, Weisbrod DJ, Eng KT, Schwartz C, Kohly R, Mandelcorn E, et al. Antibiotic resistance of ocular surface flora with repeated use of a topical antibiotic after intravitreal injection. JAMA Ophthalmol. 2013;131:456–61.
Michelle EW, Adrienne WS. How to give intravitreal injections. 2013. https://www.aao.org/eyenet/article/how-to-give-intravitre al-injections
Response: Thanks for your comments. We have revised the sentence as reviewer suggested, and related references are also cited.
“Nowadays, postoperative administration of topical antibiotics in noncomplicated cataract surgery, the most common clean surgery in ophthalmology, is a standard procedure that prevents and lowers endophthalmitis rates. However, some ophthalmologists use antibiotics in preoperative care, before the surgery, which is not proven to be beneficial for patients.[16–18] Moreover, in different ocular procedures, such as intravitreal injections, antibiotics administration is contraindicated.[19]”
2. “However, the safety of topical use of fluoroquinolones in pediatric patients are controversial.[25]”
Topical moxifloxacin, ofloxacin, ciprofloxacin and lexofloxacin are safe to be used in children older that 1 year old according to the summary of product characteristics in many countries, as in 2021.
Why in your opinion, the prescription of these drugs in children population is controversial?
Response: We mean that the safety of ocular topical use of fluoroquinolones in pediatric patients was controversial at one time, and now recent literatures considered that there was no evidence for harm of topical fluoroquinolone use in pediatrics. Thus, the prescribing of levofloxacin in pediatric patients is increasing during the study period. The revised discussion is as follows,
“However, the safety of the topical use of fluoroquinolones in pediatric patients was controversial at one time.[24] Ophthalmologists concerned about the safety of topical levofloxacin at the beginning of study and tobramycin was frequently used in pediatric patients, and the use rate of levofloxacin in pediatric patients was much lower than that in adults. Recent studies considered that there was no evidence for harm caused by topical fluoroquinolone use in pediatrics.[25–28]”
3. “Moreover, advantage of ganciclovir compared to acyclovir was not proved.”
There is a variety of ocular viral diseases caused by different types of viruses, such as adenoviruses, herpes simplex virus type-1, varicella zoster virus or cytomegalovirus.
As in human herpesviruses, there are also many forms of ocular involvement, such as conjunctivitis, blepharitis, keratitis, uveitis and retinitis. The estimated global incidence of HSV keratitis is roughly 1,5 million, including 40,000 new cases of each year. In HSV keratitis the treatment could be based on ocular and oral formulas of acyclovir or ganciclovir, without, as you wrote the advantage of ganciclovir compared to acyclovir.
However, it is not HSV, but human adenovirus (HAdV) the most common cause of infection to the ocular surface, accounting for up to 75% of conjunctivitis cases. Epidemic keratoconjunctivitis (EKC) is the most severe ocular form and is distinguished by its ability to invade the corneal epithelium, ranging in presentation from a keratitis to persistent and recurrent subepithelial infiltrates (SEIs). Possible therapeutic benefits in HAcV infection have been demonstrated through the use of antiviral drugs such as ganciclovir, ribavirin, and cidofovir.
The ophthalmic gel form of ganciclovir is used commonly in cases of HAcV conjunctivitis and keratitis. In case of this infection ganciclovir is superior to acyclovir.
Ying B, Tollefson AE, Spencer JF, et al. Ganciclovir inhibits human adenovirus replication and pathogenicity in permissive immunosuppressed Syrian hamsters. Antimicrob Agents Chemother. 2014;58(12):7171–7181. doi:10.1128/AAC.03860-14
Wold WS, Toth K. New drug on the horizon for treating adenovirus. Expert Opin Pharmacother. 2015;16(14):2095–2099. doi:10.1517/14656566.2015.1083975
Huang J, Kadonosono K, Uchio E. Antiadenoviral effects of ganciclovir in types inducing keratoconjunctivitis by quantitative polymerase chain reaction methods. Clin Ophthalmol. 2014;8:315–320. doi:10.2147/OPTH.S55284
Response: Thanks for your comments. We have revised the discussion as follows and related references are cited.
“There are a variety of ocular viral diseases caused by different types of viruses. Human adenovirus is the most common cause of infection on the ocular surface, accounting for up to 75% of conjunctivitis cases. Possible therapeutic benefits in adenovirus infection have been demonstrated through the use of antiviral drugs such as ganciclovir, ribavirin, and cidofovir. The ophthalmic formulation of ganciclovir is used commonly in cases of adenovirus conjunctivitis and keratitis, which is superior to acyclovir.[31–33] However, ganciclovir was not superior to acyclovir in ocular human herpesvirus infection, another leading cause of infectious blindness. [34]”
4. “Anti-fungal drugs were seldomly used, which was in accordance with the low prevalence of ocular fungal infection”.
The annual global incidence of fungal keratitis has never been estimated. In the recent paper published in 2020 in Lancet, the authors estimated that over a million eyes are affected each year from fungal keratitis—probably 1.4 million—assuming that in high-incidence areas culture-negative cases are usually cases of fungal keratitis. The fungal keratitis is most prevalent in tropical and subtropical locations and has been estimated to account even for 20–60% of all culture-positive corneal infections in these climates.
Thus, I am really surprised by such low use of antifungal medicines in Chine.
Response: Although the estimated incidence of fungal keratitis may be higher than we thought, the incidence of diagnosed fungal keratitis is low. Moreover, this study only includes topical formulations, and few anti-fungal drugs have available ocular topical formulations. Oral formulation and off-label use of infection formulation are always used in anti-fungal therapy. For example, voriconazole is widely used against fungal keratitis in China, but it is usually applied by dissolving voriconazole injection and dropping the solution to the eye. We have revised the sentence and add this issue into discussion.
“Moreover, oral formulations and off-label use of injectable formulations are often used, but were not included for analysis in this study.[36]”
5. What about Acanthamoeba keratitis treatment? None in China?
Response: This study has not included the diagnosis information of prescriptions into analysis. So, we cannot respond this comment.
6. Weak sides. In Europe many topical anti-infectives prescriptions are issued by general doctors (family doctors, first-line doctors), not by ophthalmologist. This could be a reason of a significant bias in your research if the situation is similar in China.
Response: Thanks for your comments. Ocular topical anti-infectives prescriptions can also be issued by general doctors as these is no restriction regarding prescribing these drugs. However, family doctors and first-line doctors are not common in China. If patients feel eye discomfort, they always come to ophthalmologist directly. The results are not affected that only prescriptions issued by ophthalmologist are included. We also added this as a limitation in the discussion.
Round 2
Reviewer 3 Report
Thank you for a prompt and appropriate response to all my queries. All suggestions were correctly addressed. I recommend the manuscript to be published in the current form.